# Tailoring the Hollow Structure within CoSn(OH)_6_ Nanocubes for Advanced Supercapacitors

**DOI:** 10.3390/molecules27227960

**Published:** 2022-11-17

**Authors:** Zhiyong Yang, Chunxia Li, Fangfang Liu, Xiaowei Lv, Lei Zhang, Yanli Fang, Hui Wang

**Affiliations:** 1Shandong Peninsula Engineering Research Center of Comprehensive Brine Utilization, School of Environment and Engineering, Weifang University of Science and Technology, Weifang 262700, China; 2State Key Laboratory Base for Eco-Chemical Engineering, College of Chemical Engineering, Qingdao University of Science and Technology, Qingdao 266042, China

**Keywords:** hollow structure, CoSn(OH)_6_, nanocubes, electrode, supercapacitor

## Abstract

The enhanced application performance of hollow-structured materials is attributed to their large surface area with more active sites. In this work, the hollow CoSn(OH)_6_ nanocubes with increased surface area and mesopores were derived from dense CoSn(OH)_6_ nanocube precursors by alkaline etching. As a result, the hollow CoSn(OH)_6_ nanocubes-based cathode electrode exhibited a higher area-specific capacitance of 85.56 µF cm^−2^ at 0.5 mA cm^−2^ and a mass-specific capacitance of 5.35 mF g^−1^ at 0.5 mA cm^−2^, which was more extensive than that of the dense precursor. Meanwhile, the current density was increased 4-fold with good rate capability for hollow CoSn(OH)_6_ nanocubes.

## 1. Introduction

Pseudocapacitors have attracted extensive research interests with belief in their merits such as high capacitance, fast rechargeable ability, and safety. From the literature point of view, the active component on the electrodes has played a vital role in the performance of pseudocapacitors [1,2,3]. Transition metal oxides have been deemed the most widely studied materials at present due to their merits such as large theoretical capacity, rich reserves, and non-toxic characteristics. Furthermore, in the pursuit of higher stability and conductivity of the capacitor electrode, researchers began to mix the hydro/oxide of binary metals instead of a single metal’s hydro/oxide. Many kinds of hydro/oxides of binary metals have been reported, such as NiCo- [4,5,6] and CoSn [7,8,9]-based hydro/oxides, and these materials were also used in other electrochemical applications such as electro-catalysts, lithium-ion batteries, and supercapacitors [10,11]. For instance, Song et al. [12] prepared CoSn(OH)_6_ nanocubes via the electrochemical etching approach. As the catalysts for the oxygen reduction reaction, the over-potential of the oxygen reduction reaction was only 274 mV at 10 mA cm^−2^. Wang et al. [8] prepared amorphous CoSnO_3_@C nano-boxes using box-like CoSn(OH)_6_ crystalline, which can endure 400 cycles in lithium-ion batteries. The reports regarding the capacitor application are few, although CoSn(OH)_6_ nanocubes exhibited good capacitor behavior in the studies by Wang et al. [9]. However, the specific abilities of these transition metal compounds were usually reported to be smaller in practice than their capacities in theory. This could be ascribed to the limited exposure of the active sites within the structure of the active component that otherwise would be fully utilized. Thus, the increased exposure of more active areas of transition metal compounds has been addressed by various strategies such as changing the morphology, enhancing the dispersion such as by loading NiCo_2_S_4_ on the surface of carbon nanotubes [13], creating pores, and so on [14,15,16,17].

Out of these strategies, the creation of hollow structures with various morphologies holds promise in the creation of more active sites due to the high-surface-volume ratio feature and could sequentially achieve higher pseudocapacitance [18,19]. For instance, Liang and co-workers [20] reported hollow NiCo_2_S_4_ spheres with a specific capacitance of 756 F g^−1^ at 1 A g^−1^, and the hollow CoMn_2_O_4_ spheres fabricated by Hussain et al. [21] exhibited 168 mA h^−1^ at a current density of 1 A g^−1^. In the work of Sudheendra Budhiraju et al. [22], the hollow NiMoO_4_ fibers delivered a capacity of 214 mAh g^−1^ at the current density of 2 A g^−1^. Wang et al. [23] prepared hollow-structure NiCoP nanorods via the Kirkendall effect, and a capacitance of 273.4 μAh cm^−2^ was achieved at the current density of 30 mA cm^−2^ with a retention rate of 85.6%. A cubic double-layer structured Cu_7_S_4_/NiS composite was synthesized by Cao and co-workers [24] using a self-generated sacrificial template method, and the composite delivered a specific capacitance of 1028 F g^−1^ at a 1 A g^−1^ current density. Moreover, this strategy was also demonstrated by hierarchical nickel sulfide (NiS_x_) hollow microspheres [25], hollow microspheres of Ni-Co layered double hydroxides (LDHs), and Ni-Mn LDHs [26]. However, the deviation between their theoretical capacities and actual performance has posed an interesting research question regarding a deeper structural understanding. In this work, hollow CoSn(OH)_6_ nanocubes were derived from dense CoSn(OH)_6_ nanocube precursors by applying an alkaline etching approach. The results showed that the hollow structure exhibited a 4-fold increase in current density compared to that of the dense CoSn(OH)_6_ precursor.

## 2. Results and Discussion

### 2.1. Physical Characterization

Figure 1 shows the XRD patterns and SEM and TEM images of the prepared samples. As seen in Figure 1a, the diffraction peaks are in agreement with the data of the standard CoSn(OH)_6_ card (PDF# 13-0356). As observed from the SEM image in Figure 1b, the CoSn(OH)_6_ sample exhibits a dense cube-like shape with an edge length of 120 nm. After the etching process in an alkaline solution, the cube-like shape was retained while a hollow structure formed with certain facets missing from the cubes (Figure 1c). The TEM image in Figure 1d further demonstrates the hollow structure of CoSn(OH)_6_ nanocubes, though certain nanocubes are not perfectly hollow. This non-uniformity could be caused by the nonideal etching reaction. The inset in Figure 1d shows the EDS result of hollow CoSn(OH)_6_ nanocubes, which further implies the formation of CoSn(OH)_6_ via the presence of Co and Sn elements.

The chemical structure of hollow CoSn(OH)_6_ samples was studied using infrared spectroscopy and the results are shown in Figure 2 [27]. The wideband at 3423 cm^−1^ and the small peak at 1637 cm^−1^ represent the tensile and bending vibrations of the O-H bonds, respectively. The Sn-O bond’s stretching vibration and the Sn-OH bond’s bending vibration correspond to approximately 538 cm^−1^ and 1177 cm^−1^. The strong band at 786 cm^−1^ represents the H_2_O-H_2_O hydrogen bond.

Figure 3 shows the surface composition of the hollow CoSn(OH)_6_ samples [10]. As shown in Figure 3a, the wide-range XPS measurement spectrum confirmed that the synthesized hollow CoSn (OH)_6_ material is composed of Co, Sn, and O elements, which is consistent with the EDX test results. Figure 3b is a high-resolution XPS spectrogram of Co 2p. The two peaks at approximately 778.0 eV and 794.0 eV correspond to Co 2p_3/2_ and Co 2p_1/2_, respectively. The satellite peaks located at 780.4 and 799.8 eV are usually caused by multi-electron excitation (pulse) or coupling between unpaired electrons (multistate splitting). In the high-resolution XPS spectrum of Sn 3d (Figure 3c), the high-intensity peak with a binding energy of 486.6 eV is Sn 3d_5/2_ and the peak at 495.1 eV is Sn 3d_3/2_. The peak of Sn 3d splits at 8.5 eV, indicating that Sn is in the chemical state of Sn (IV). The high-resolution XPS spectra of O 1s can be divided into two photoelectron peaks at 531.3 and 533.1 eV, considered lattice oxygen in hydroxide structure and oxygen in hydroxide ions, respectively (Figure 3d) [11].

Further evidence of the formation of the hollow structure is the change in specific surface areas. As shown in Figure 4, N_2_ isotherms of both samples present the feature of mixed type I/IV curves classified by the international union of pure and applied chemistry (IUPAC). Clearly, the presence of micropores in two samples can be reflected by the apparent uptake over the low-pressure region, and the hysteresis loop spread across the high-pressure region suggests the existence of mesopores. The specific surface area of the hollow nanocubes is 34.7 m^2^ g^−1^ as calculated by the Brunauer–Emmett–Teller method, which is larger than that of the dense nanocubes with a specific surface area of 9.3 m^2^ g^−1^. The increase in the specific surface area for these nanocubes results from the etching. The pore size for dense CoSn(OH)_6_ nanocubes is scattered as seen in the inset of Figure 4a, and at the same time, the pore size of hollow CoSn(OH)_6_ nanocubes is concentrated at ca. 30 nm, which could be ascribed to the etching and demonstrates the formation of the hollow CoSn(OH)_6_ structure.

### 2.2. Electrochemical Measurements

Cyclic voltammetry (CV) [28] and the constant current charge–discharge (GCD) were used to study the electrochemical performance of the prepared hollow CoSn(OH)_6_ nanocubes-based electrode in the conventional three-electrode system, and its performance was compared with that of dense CoSn(OH)_6_ and the foam nickel substrate (NF). Figure 5a shows the CV curves of all three at a current density of 5 mV s^−1^. At a glance, the intensity of the current density on the nickel foam support electrode is weak compared to those of the other electrodes, indicating that the contribution of the nickel foam support to the total capacity is small. It can be seen from the figure that the response current of the hollow CoSn(OH)_6_ nanocubes-based electrode is more obvious than that of the dense CoSn(OH)_6_ nanocubes-based electrode, suggesting that CoSn(OH)_6_ nanocubes with hollow structures have higher charge-storage capacity than the dense CoSn(OH)_6_ nanocubes, which is inseparable from the fact that hollow structures can provide greater specific surface area and more active sites. This result shows that the performance of CoSn(OH)_6_ is significantly improved. Similarly, the GCD result shown in Figure 5b is consistent with the conclusion obtained by CV, which further proves this conclusion. In addition to the better performance of the hollow structure according to the size of the area enclosed by the CV curve, it can also be seen that the CV curves of both dense and hollow materials contain obvious depolarization, which is due to the redox conversion between Co (II) and Co (III) [29], and the appearance of this redox peak also represents the contribution of the pseudocapacitance, which corresponds to the nonlinear GCD curve. This indicates the pseudocapacitance behavior of the CoSn(OH)_6_ nanocubes material [28].

The rate ability is one of the crucial indicators to evaluate the advantages of pseudocapacitor materials. GCD curves at different current densities were used to test the rate abilities of these nanocubes. As illustrated in Figure 5c, the GCD curves of hollow CoSn(OH)_6_ nanocubes retain a similar shape all the way, and the discharge time decreases with the increased current density. We calculated the area-specific capacity according to the formula $C_{a}=I*t/\left(A*\Delta V\right)$ (where I, t, ΔV, and A are the discharge current (A), discharge time (s), potential window (ΔV), and electrode area (cm^2^), respectively) [30]. The corresponding specific capacity was calculated and plotted in Figure 5d to illustrate the connection of the individual capacity with the current density of dense CoSn(OH)_6_ nanocubes. The specific capacity of hollow CoSn(OH)_6_ nanocubes is 16.17 µA h cm^−2^ and its retention is 38% as the current density increases 4-fold. Meanwhile, the retention of dense CoSn(OH)_6_ nanocubes is 41.7% under the same change in current density. The results indicate that the hollow CoSn(OH)_6_ structure did not improve the rate ability of these nanocubes, and this could be ascribed to the unimproved electric conductivity. Thus, enhancing the electric conductivity of the electrode could be another strategy to allow CoSn(OH)_6_ nanocubes to be applied in pseudocapacitors. Electrochemical impedance also confirms that hollow CoSn(OH)_6_ nanocubes h better rate performance. In the impedance comparison of the two electrodes, the hollow CoSn(OH)_6_ nanocubes electrode has the smallest resistance value during the electrode redox energy storage process (Figure 5e). A stability test conducted on hollow CoSn(OH)_6_ nanocubes exhibits cycling stability, with a capacity retention rate of 40% after 10,000 cycles (Figure 5f).

To evaluate the performance of the hollow CoSn(OH)_6_ nanocubes electrode in practical applications, the symmetric capacitor was assembled, and the electrochemical performance of the electrode in a two-electrode system was further studied [11]. In order to determine the appropriate operating voltage window range of the symmetrical capacitor, CV tests are carried out at different voltages starting from 0 V at a scanning speed of 100 mV s^−1^. As can be seen from Figure 6a, the assembled symmetrical capacitor exhibits obvious polarization in the 1.2 V voltage window. Therefore, 0–1.1 V is determined to be the appropriate operating voltage range. Furthermore, in Figure 6b, as the charge and discharge current densities increase from 5 to 20 mA cm^−2^, the GCD curve of the hollow CoSn(OH)_6_ nanocubes electrode still maintains a suitable line type, indicating that the electrode has an excellent rate performance. When the current density is 5 mA cm^−2^, the specific capacity of the symmetrical capacitor is 7.94 μAh cm^−2^. When the current density increases from 5 to 20 mA cm^−2^ (Figure 6c), the capacity decreases to 5.45 μAh cm^−2^, which shows that the capacitor has good rate performance. To further study the electrochemical performance of the symmetric capacitor, EIS was carried out. Figure 6d shows the Nyquist diagram of a capacitor composed of hollow CoSn(OH)_6_ nanocubes electrodes before and after 10,000 charge–discharge cycles. The curve in the figure is composed of a semicircle in the high-frequency region and a straight line in the low-frequency region from which Rs representing the ohmic internal resistance and Rct representing the charge transfer resistance can be obtained. Through fitting, the Rs and Rct values of the two curves before and after the cycle are obtained, in which the Rs difference is 0.29 Ω and Rct difference is 2.2 Ω. These values indicate the electrochemical stability of the material, i.e., the CoSn(OH)_6_//AC asymmetric capacitor has cycle durability [10]. All of these results suggest that the electrode conductivity is high, and this accounts for the good electrochemical performance of the symmetric capacitor.

## 3. Materials and Methods

### 3.1. Synthesis of Hollow CoSn(OH)_6_ Nanocubes 

All the chemicals used in the synthesis of hollow CoSn(OH)_6_ nanocubes were analytical grades. In a typical experiment, 3 mmol of CoCl_2_∙H_2_O together with 3 mmol of sodium citrate were dissolved in 115 mL of deionized water in a beaker. Then, the obtained solution was transferred to a 250 mL flask and mixed thoroughly with 15 mL of a 0.2 mol L^−1^ SnCl_4_/ethanol solution under 5 min stirring. Sequentially, the mixture was kept stirring for another 1 h to allow the Sn^4+^_(aq)_ + Co^2+^_(aq)_ + 6OH^−^_(aq)_ → CoSn(OH)_6(s)_ reaction to take place. After the reaction, the pink product was collected, washed with deionized water/ethanol, and dried at 60 °C under a vacuum to obtain the dense CoSn(OH)_6_ nanocubes sample. 

Afterwards, the hollow CoSn(OH)_6_ nanocubes were synthesized based on the 4CoSn(OH)_6_ + 4 OH^−^_(aq)_ → [Co(OH)_4_]^2−^_(aq)_ + [Sn(OH)_6_]^2−^_(aq)_ reaction by immersing the dense CoSn(OH)_6_ nanocubes sample into 60 mL of an 8 mol L^−1^ NaOH solution for 20 h under stirring.

### 3.2. Characterization

The morphologies of the synthesized samples were examined on a Carl Zeiss Ultra Plus scanning electron microscopy (SEM), and their crystallographic structures were identified on an X-ray diffractor (XRD, Shimadzu XD–3A, Tokyo, Japan), which used filtered Cu-Kα radiation and operated at 40 kV and 30 mA. The internal structures of the samples were imaged on a transmission electron micrograph (TEM, JEOL, JEM-2000 FX) that was operated at 200 kV, and coupled energy dispersive spectroscopy (EDS) was used for the elemental analysis. The surface area and pore size distribution were measured on a Quantachrome Instrument (autosorb IQ) using N_2_ as the working gas. The functional groups present on the sample surfaces were studied by Fourier Transform infrared spectroscopy (FT-IR) recorded on a PerkineElmer spectrophotometer (FT-IR M-1700 2B). X-ray photoelectron spectroscopy (XPS) characterizations were carried out on a PHI Quantum 2000 XPS system with a monochromatic Al Kα source and a charge neutralizer. The X-ray photoelectron spectra of all the elements were referenced as the C 1s peak arising from adventitious carbon (its binding energy was set at 284.8 eV).

### 3.3. Electrochemical Measurements 

The cyclic voltammetry (CV) curves were recorded on a three-electrode configured electrochemical workstation (CHI 660E). Typically, the working electrode (1 × 1 cm^2^) was prepared by pressing the active components of CoSn(OH)_6_, carbon black, and PTFE with nickel foam in a mass ratio of 8:1:1, and the overall mass loading on the electrode was 3.1 mg cm^−2^. Similarly, the counter electrode was fabricated using active carbon (AC) as the active component. Hg/HgO (1.0 M KOH) was applied as the reference electrode, and the aqueous electrolyte (1 M KOH) was used as the electrolyte. The galvanostatic charging/discharging (GCD) process was conducted on an instrument (LAND CT2001A). The measurement range of the electrochemical impedance spectrum was 1.0 × 10^−2^ to 1.0 × 10^6^ Hz, and the voltage selected was open-circuit voltage.

## 4. Conclusions

Hollow-structured CoSn(OH)_6_ nanocubes were synthesized using dense nanocubes via alkaline etching. Benefitting from the enhanced surface area and large mesopores, a pesudocapacitor cathode based on hollow CoSn(OH)_6_ nanocubes displayed an area-specific capacitance of 85.56 µF cm^−2^ at 0.5 mA cm^−2^ and a mass-specific capacitance of 5.35 mF g^−1^ at 0.5 mA cm^−2^. In contrast, the performance was higher than that of dense CoSn(OH)_6_ nanocubes. On the other hand, hollow CoSn(OH)_6_ nanocubes did not show a better rate capability than the dense sample because of their limited electric conductivity. Indeed, electrical conductivity needs to be improved when applying hollow CoSn(OH)_6_ nanocubes as the active component for pseudocapacitors.

## Figures and Tables

**Figure 1 molecules-27-07960-f001:**
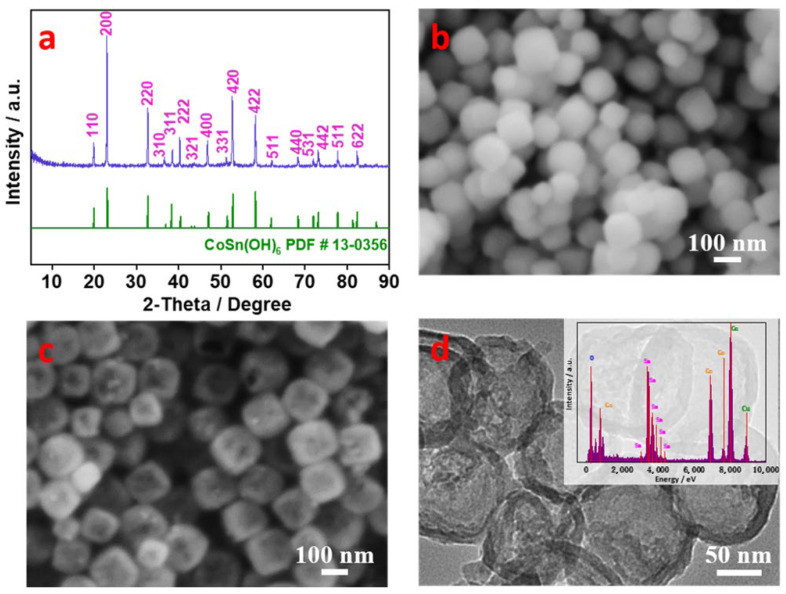
(**a**) XRD pattern of CoSn(OH)_6_ nanocubes; SEM images of (**b**) dense and (**c**) hollow CoSn(OH)_6_ nanocubes; (**d**) TEM image of hollow CoSn(OH)_6_ nanocubes (Inset: EDS analysis).

**Figure 2 molecules-27-07960-f002:**
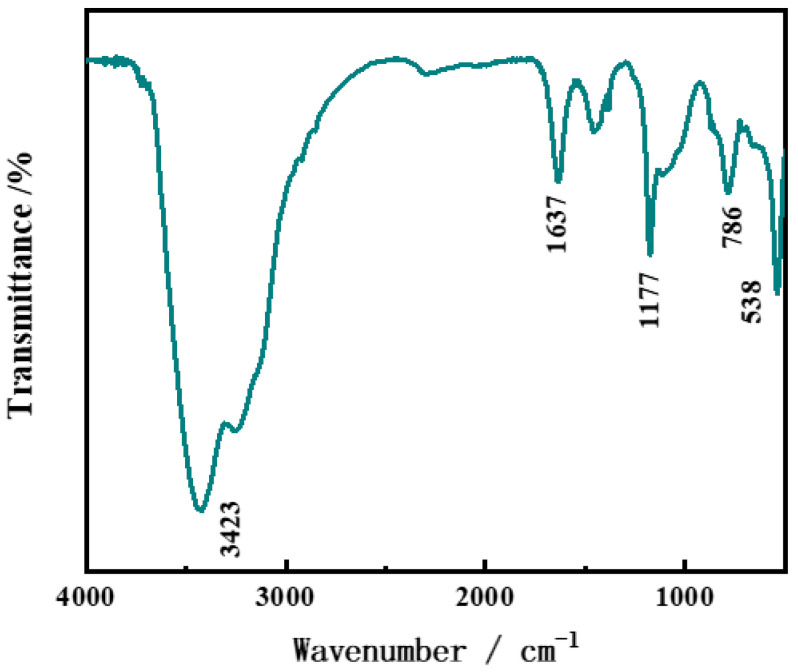
FTIR spectrum of hollow CoSn(OH)_6_.

**Figure 3 molecules-27-07960-f003:**
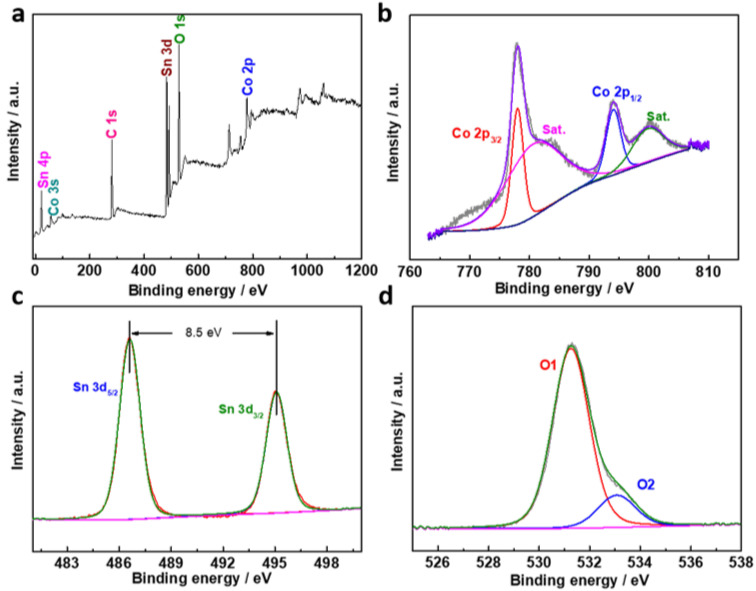
XPS spectra of hollow CoSn(OH)_6_ sample. (**a**) Survey spectrum, (**b**) Co 2p, (**c**) Sn 3d, and (**d**) O 1s high-resolution XPS spectra.

**Figure 4 molecules-27-07960-f004:**
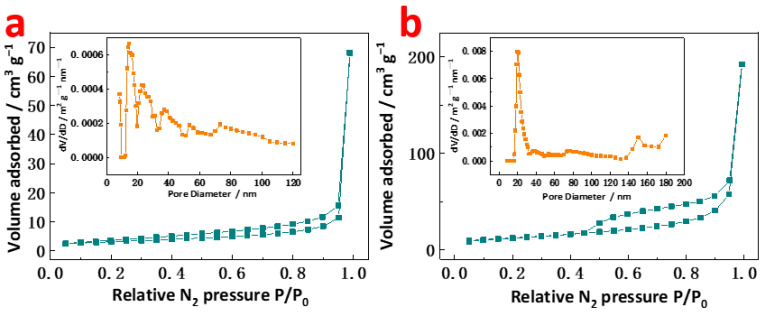
N_2_ isotherms and corresponding pore size distribution of (**a**) dense and (**b**) hollow CoSn(OH)_6_ nanocubes.

**Figure 5 molecules-27-07960-f005:**
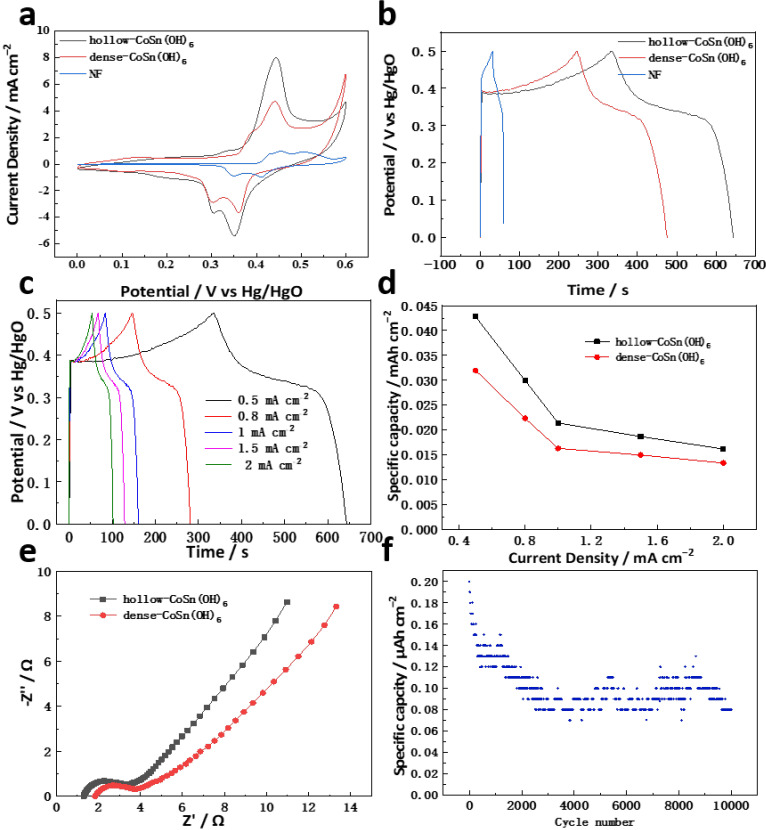
(**a**) CV curves at scan rate of 5 mV s^−1^ and (**b**) GCD curves at current density of 0.5 mA cm^−2^ of dense and hollow CoSn(OH)_6_ nanocubes. (**c**) GCD curves of hollow CoSn(OH)_6_ nanocubes and (**d**) corresponding variation in area-specific capacity at different current densities of dense and hollow CoSn(OH)_6_ nanocubes. (**e**) Electrochemical impedance spectroscopy of dense and hollow CoSn(OH)_6_ nanocubes. (**f**) Cycle stability of hollow CoSn(OH)_6_ nanocubes.

**Figure 6 molecules-27-07960-f006:**
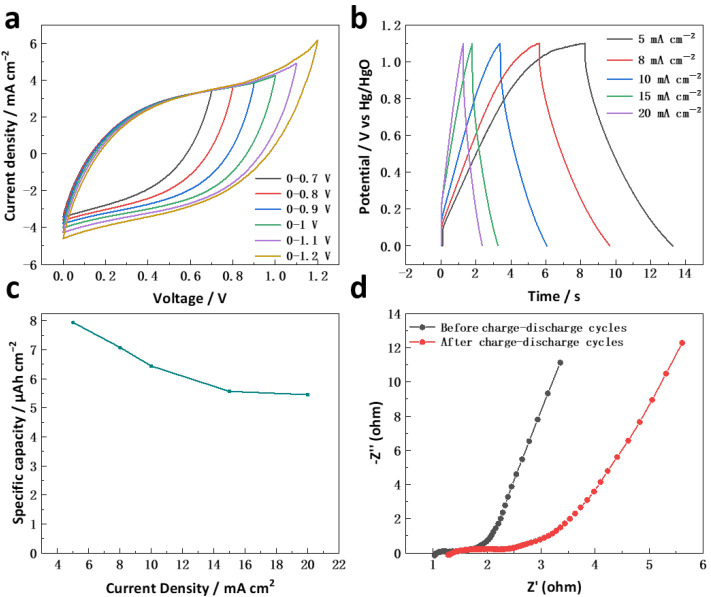
(**a**) CV curves measured at different operating voltages at 100 mV s^−1^, (**b**) GCD curves collected over different current densities, (**c**) capacities vs. current densities, (**d**) Nyquist plot of the same before and after 10,000 charge–discharge cycles.

## Data Availability

Data can be made available upon written request to the corresponding author and with proper justification.

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
