# Peer review of "Tailoring the Hollow Structure within CoSn(OH)_6_ Nanocubes for Advanced Supercapacitors"

_molecules, 2022, doi:10.3390/molecules27227960_

Round 1
Reviewer 1 Report
Page 3, line 70-74. It is necessary to support the FTIR results with the bibliography.
Figure 2. Generally, FTIR patterns range from highest to lowest wavenumber.
In the XPS methodology, the criteria used in peak deconvolution are not specified.
Page 3-4, line 78-89. It is necessary to support the XPS results with the bibliography.
Page 4. It is necessary to support the results of textural analysis with the bibliography.
Page 5. ….. “As shown in Figure 5a, the CVs of dense- and hollow- CoSn(OH)6 nanocubes display a couple of redox peaks related to the transition of Co2+/Co3+”…… Shown in figure 5a.
In the electrochemical methodology, the amount of material used in the measurements is not indicated. This is fundamental because the values obtained can be overestimated or underestimated. Also, the experimental conditions under which the electrochemical impedance spectroscopy was performed are not specified and the proposed equivalent circuit is not supported either. There is no heat if the assembly of two electrodes is carried out because it is not described in the methodology
The author confuses on several occasions the terms of specific capacitance with specific capacity, so it is necessary to review the concepts and base the discussion of his results.
It is necessary to corroborate that it is a pseudocapacitive process. For this, it is necessary to review the following bibliography.
a) Roldan, S.; Barreda, D.; Granda, M.; Menendez, R.; Santamaria, R.; Blanco, C., An approach to classification and capacitance expressions in electrochemical capacitors technology. Physical Chemistry Chemical Physics 2015, 17 (2), 1084-1092.;
b) Brisse, A.-L.; Stevens, P.; Gwenaelle, T.; Crosnier, O.; Brousse, T., Ni(OH)2 and NiO Based Composites: Battery Type Electrode Materials for Hybrid Supercapacitor Devices. 2018; Vol. 11, p 1178.
c) Balducci, A.; Belanger, D.; Brousse, T.; Long, J. W.; Sugimoto, W., Perspective—A Guideline for Reporting Performance Metrics with Electrochemical Capacitors: From Electrode Materials to Full Devices. Journal of The Electrochemical Society 2017, 164 (7), A1487-A1488.
d) Laheäär, A.; Przygocki, P.; Abbas, Q.; Béguin, F., Appropriate methods for evaluating the efficiency and capacitive behavior of different types of supercapacitors. Electrochemistry Communications 2015, 60, 21-25.
e) T. Brousse, D. Bélanger, J. W. Long, Journal of The Electrochemical Society 2015, 162, A5185-A5189 10.1149/2.0201505jes.
Page 6 It is necessary to support the results of electrochemical performance with the bibliography
Author Response
Dear Editor,
Referring to the manuscript titled “Tailoring the Hollow Structure within CoSn(OH)6 Nanocubes for Advanced Supercapacitors”, we thank all the reviewers for their valuable comments. The point-to-point response/revision can be found below:
Reviewer #1: Comments and Suggestions for Authors:
Comments 1: Page 3, line 70-74. It is necessary to support the FTIR results with the bibliography.
Response/revision: Thanks for this suggestion. The research work below has been
cited as reference:
- Lin, X.; Gao, Y.; Jiang, M.; Zhang, Y.; Hou, Y.; Dai, W.; Wang, S.; Ding, Z., Photocatalytic CO2 reduction promoted by uniform perovskite hydroxide CoSn(OH)6 nanocubes. Applied Catalysis B: Environmental 2018, 224, 1009-1016.
Comments 2: Figure 2. Generally, FTIR patterns range from highest to lowest wavenumber.
Response/revision: Thank you for your suggestion. Figure 2 has been modified and reflected in the text.
Comments 3: In the XPS methodology, the criteria used in peak deconvolution are not specified.
Response/revision: Thank you for your question. After the background of XPS test characterization analysis is eliminated, Shirley method is used for deconvolution, and Gaussian Lorentz method is used for deconvolution at the peak value. We are described XPS characterization in "3.2 Characterization":X-ray photoelectron spectroscopy (XPS) characterizations were carried out on a PHI Quantum 2000 XPS system with a monochromatic Al Kα source and a charge neutralizer. The X-ray photoelectron spectra of all of the elements were referenced to the C 1s peak arising from adventitious carbon (its binding energy was set at 284.8 eV).
Comments 4: Page 3-4, line 78-89. It is necessary to support the XPS results with the bibliography.
Response/revision: Thanks for this suggestion. The research work below have been
cited as reference:
- Sahoo, R.; Sasmal, A. K.; Ray, C.; Dutta, S.; Pal, A.; Pal, T., Suitable Morphology Makes CoSn(OH)6 Nanostructure a Superior Electrochemical Pseudocapacitor. ACS Appl Mater Interfaces 2016, 8, (28), 17987-98.
Comments 5: Page 4. It is necessary to support the results of textural analysis with the bibliography.
Response/revision: Thank you for your advice. Page 4 is an analysis of XPS. Reference 27 is used as a reference:
- Li, B.; Zhang, G.-X.; Huang, K.-S.; Qiao, L.-F.; Pang, H., One-step synthesis of CoSn(OH)6 nanocubes for high-performance all solid-state flexible supercapacitors. Rare Metals 2017, 36, (5), 457-464
Comments 6: Page 5. ….. “As shown in Figure 5a, the CVs of dense- and hollow- CoSn(OH)6 nanocubes display a couple of redox peaks related to the transition of Co2+/Co3+”…… Shown in figure 5a. In the electrochemical methodology, the amount of material used in the measurements is not indicated. This is fundamental because the values obtained can be overestimated or underestimated. Also, the experimental conditions under which the electrochemical impedance spectroscopy was performed are not specified and the proposed equivalent circuit is not supported either. There is no heat if the assembly of two electrodes is carried out because it is not described in the methodology.
Response/revision: Thank you for your suggestion. The following is a description of the test conditions used in this experiment: CoSn(OH)6, carbon black, and polytetrafluoroethylene are mixed and pressed onto foam nickel at a mass ratio of 8:1:1, then cut into 1×1 cm2 for the working electrode. The total mass of the load material on the electrode is 3.1 mg cm-2.
For the electrochemical impedance test, the selected measurement range is 10-2 to 106 Hz, and the open circuit voltage is selected as the voltage. The equivalent circuit is selected by directly importing the impedance test data into the ZSmDemo software. It may be deficient according to the closest degree between the fitting curve and the measured curve, so we remove it from Figure 6.
Comments 7: The author confuses on several occasions the terms of specific capacitance with specific capacity, so it is necessary to review the concepts and base the discussion of his results.
It is necessary to corroborate that it is a pseudocapacitive process. For this, it is necessary to review the following bibliography.
- a) Roldan, S.; Barreda, D.; Granda, M.; Menendez, R.; Santamaria, R.; Blanco, C., An approach to classification and capacitance expressions in electrochemical capacitors technology. Physical Chemistry Chemical Physics 2015, 17 (2), 1084-1092.;
- b) Brisse, A.-L.; Stevens, P.; Gwenaelle, T.; Crosnier, O.; Brousse, T., Ni(OH)2 and NiO Based Composites: Battery Type Electrode Materials for Hybrid Supercapacitor Devices. 2018; Vol. 11, p 1178.
- c) Balducci, A.; Belanger, D.; Brousse, T.; Long, J. W.; Sugimoto, W., Perspective—A Guideline for Reporting Performance Metrics with Electrochemical Capacitors: From Electrode Materials to Full Devices. Journal of The Electrochemical Society 2017, 164 (7), A1487-A1488.
- d) Laheäär, A.; Przygocki, P.; Abbas, Q.; Béguin, F., Appropriate methods for evaluating the efficiency and capacitive behavior of different types of supercapacitors. Electrochemistry Communications 2015,60, 21-25.
- e) T. Brousse, D. Bélanger, J. W. Long, Journal of The Electrochemical Society 2015, 162, A5185-A5189 10.1149/2.0201505jes.
Response/revision: Thank you for your recommendation. After reading the literature you provided, I have a certain understanding of the difference between "specific capacitance" and "specific capacity". I have revised the relevant statements in the text.
Comments 8: Page 6 It is necessary to support the results of electrochemical performance with the bibliography
Response/revision: Thanks for this suggestion. The research work below have been
cited as reference:
- Li, B.; Zhang, G.-X.; Huang, K.-S.; Qiao, L.-F.; Pang, H., One-step synthesis of CoSn(OH)6 nanocubes for high-performance all solid-state flexible supercapacitors. Rare Metals 2017, 36, (5), 457-464.

Reviewer 3 Report
Manuscript ID: molecules-1992213
Manuscript title: Tailoring the Hollow Structure within CoSn(OH)6 Nanocubes for Advanced Supercapacitors
The research paper concerns the preparation, characterization, and application of a hollow structure CoSn(OH)6 materials of potential importance in electrochemical capacitors. Below the Authors, and the Editor can find my comments and questions regarding the manuscript:
- Page 2, line 50: lettering in word “approach”. Please check the manuscript carefully in terms of lettering, punctuation, and grammar. Especially the electrochemical part, in my opinion, is rather poorly written. The language and style must be certainly improved.
- Fig. 1d – EDX spectra: I think the authors meant Co, not Cu in the description of the EDX signal.
- Fig. 5d, and 6c,Y axis: I recommend changing “capacitance” into “capacity”, and avoiding the former in the description of the material properties in the three-electrode cell (see details below).
- The N2 adsorption/desorption results are not very informative. It is necessary to add at least the calculated value of BET surface area and total pore volume of the samples.
- The charge storage contribution from the Ni foam current collector should be included or at least discussed.
- The mechanism of formation: is any synergistic effect between Co- and Sn- expected (when it comes to the mutual interaction, eg. changes in the electron densities) with respect to the single Co-, or Sn-based materials or it is a rather purely physical mixture of the components (and was it investigated)? I wonder what are the benefits of using Co/Sn admixture because it is barely clear from the paper.
- I don’t understand why the charging voltage of a symmetric capacitor (Fig. 6) is set between 0.2 and 0.8-1.1 V. It has no justification to me. Please record and show data between 0 V and the upper voltage limit.
- I found no information about the calculations of the specific capacities (capacitances) in both the three- and the two-electrode configuration. Meanwhile, the specific (gravimetric) capacities shown are extremely low, i.e. on the level of microfarads or micro-Amper-h). It is hard to consider such material for energy storage at all. Interestingly, judging from the GCD curves and given information, such a result seems to be hugely underestimated. However, it is hard to verify it because some important data is missing (geometric area of the electrodes, calculations procedure).
- Please be careful with the term “pseudocapacitance” or “pseudocapacitive material” in the case of the investigated composite. According to recent lively discussions, the material under study may, or may not, have pseudocapacitive nature - that should be verified at first. It is clear, however, that its charge storage properties (in the three-electrode cell) can be expressed as capacitance (in farads), in addition to the advised unit of Ah (such as for battery-type materials), only under certain conditions (i.e. by applying specific calculation procedures, here the information about applied procedures is missing). On contrary, the full symmetric cell characteristics presented in the manuscript can be classified as a capacitor-like device (with a capacitance in farads as the parameter describing its charge storage abilities) due to its specific (capacitive-like) electrochemical signature (i.e. changes of charge as a function of voltage are almost linear, hence the capacitance, if determined as C=Q/U, will be more or less constant over the whole voltage window). Please analyze the following papers (especially the last one) to understand the problem more deeply. Additionally, I noticed a reference (no 21) to the first paper I also advise. The conclusions formulated by prof. Brousse et al. are, however, in my opinion, different than suggested by the Authors.
DOI: 10.1149/2.0201505jes
DOI: 10.1016/j.elecom.2015.07.022
https://doi.org/10.1021/acs.chemrev.0c00170
To conclude, in my opinion, the paper cannot be published in Molecules in its current form. It needs careful revision and clarification of the presented data. It can be reconsidered after major revision.
Author Response
Dear Editor,
Referring to the manuscript titled “Tailoring the Hollow Structure within CoSn(OH)6 Nanocubes for Advanced Supercapacitors”, we thank all the reviewers for their valuable comments. The point-to-point response/revision can be found below:
Reviewer #3::The research paper concerns the preparation, characterization, and application of a hollow structure CoSn(OH)6 materials of potential importance in electrochemical capacitors. Below the Authors, and the Editor can find my comments and questions regarding the manuscript:
Comments 1: Page 2, line 50: lettering in word “approach”. Please check the manuscript carefully in terms of lettering, punctuation, and grammar. Especially the electrochemical part, in my opinion, is rather poorly written. The language and style must be certainly improved.
Response/revision: Thank you for your advice. Now, some modifications have been made to the electrochemical part. See the text for details.
Comments 2: Fig. 1d – EDX spectra: I think the authors meant Co, not Cu in the description of the EDX signal.
Response/revision: Thank you for your question. As a carrier network, Cu will show signals in EDS. We have re characterized the samples by EDS, analyzed them, and re labeled the elements such as Co and Sn. See the text for the modified Figure 1d.
Comments 3: Fig. 5d, and 6c,Y axis: I recommend changing “capacitance” into “capacity”, and avoiding the former in the description of the material properties in the three-electrode cell (see details below).
Response/revision: Thank you for your advice. The ordinates of the two figures have been modified. See the text for the modified figure.
Comments 4: The N2 adsorption/desorption results are not very informative. It is necessary to add at least the calculated value of BET surface area and total pore volume of the samples.
Response/revision: Thank you for your advice. The N2 adsorption desorption curve was analyzed and described again. See the text for the revised content.
Comments 5: The charge storage contribution from the Ni foam current collector should be included or at least discussed.
Response/revision: Thank you for your advice. We separately tested the CV curve of foam nickel (NF) substrate under different scanning speeds and the GCD curve under different current densities in the three electrode system, and the results are shown in the following figure:
It can be seen from the figure that NF still shows obvious pseudo capacitance behavior, and it is compared with the performance of hollow/dense CoSn(OH)6 nano cube. See Figure 5 (a, b) in the text for the comparison figure. It can be seen that the performance of NF alone is poor, and the performance is significantly improved after loading active substances. See the text for relevant description.
Comments 6: The mechanism of formation: is any synergistic effect between Co- and Sn- expected (when it comes to the mutual interaction, eg. changes in the electron densities) with respect to the single Co-, or Sn-based materials or it is a rather purely physical mixture of the components (and was it investigated)? I wonder what are the benefits of using Co/Sn admixture because it is barely clear from the paper.
Response/revision: Thank you for your advice. The advantages of using Co/Sn mixture are described in the foreword and proved by citing certain references. See the text for the revised contents.
Comments 7: I don’t understand why the charging voltage of a symmetric capacitor (Fig. 6) is set between 0.2 and 0.8-1.1 V. It has no justification to me. Please record and show data between 0 V and the upper voltage limit.
Response/revision: Thank you for your suggestion. Now, CV and GCD tests are conducted again with 0 V as the starting point. See Figure 6 and the text for the test analysis results.
Comments 8: I found no information about the calculations of the specific capacities (capacitances) in both the three- and the two-electrode configuration. Meanwhile, the specific (gravimetric) capacities shown are extremely low, i.e. on the level of microfarads or micro-Amper-h). It is hard to consider such material for energy storage at all. Interestingly, judging from the GCD curves and given information, such a result seems to be hugely underestimated. However, it is hard to verify it because some important data is missing (geometric area of the electrodes, calculations procedure).
Response/revision: Thank you for your question. In this experiment, the mixed transition metal hydroxides are used in electrode materials because they have stronger electrochemical activity than single metal hydroxides, but sometimes they also show reduced specific capacitance, which may be due to the lack of participation of all. Therefore, this experiment shortens the ion diffusion path by creating a hollow nano-shell structure, providing a larger specific surface area, and the generated internal cavity can reduce the stress, improve the bearing capacity, so as to load more active substances. This is consistent with the purpose of other researchers to solve this limitation by introducing pores or creating porous layered structures. In a word, this experiment emphasizes this preparation method, not just its performance in capacitors.
Comments 9: Please be careful with the term “pseudocapacitance” or “pseudocapacitive material” in the case of the investigated composite. According to recent lively discussions, the material under study may, or may not, have pseudocapacitive nature - that should be verified at first. It is clear, however, that its charge storage properties (in the three-electrode cell) can be expressed as capacitance (in farads), in addition to the advised unit of Ah (such as for battery-type materials), only under certain conditions (i.e. by applying specific calculation procedures, here the information about applied procedures is missing). On contrary, the full symmetric cell characteristics presented in the manuscript can be classified as a capacitor-like device (with a capacitance in farads as the parameter describing its charge storage abilities) due to its specific (capacitive-like) electrochemical signature (i.e. changes of charge as a function of voltage are almost linear, hence the capacitance, if determined as C=Q/U, will be more or less constant over the whole voltage window). Please analyze the following papers (especially the last one) to understand the problem more deeply. Additionally, I noticed a reference (no 21) to the first paper I also advise. The conclusions formulated by prof. Brousse et al. are, however, in my opinion, different than suggested by the Authors.
DOI: 10.1149/2.0201505jes
DOI: 10.1016/j.elecom.2015.07.022
https://doi.org/10.1021/acs.chemrev.0c00170
Response/revision: Thank you for your suggestions and recommendations. I carefully read the three articles you provided, and have a deeper understanding of the properties of pseudo capacitors. Pseudo capacitance is between capacitive behavior and Faradaic behavior, but there are many factors that affect its storage mechanism to determine which behavior it belongs to. For example, the nature and morphology of materials are very important. Some materials (such as MnO2, RuO2, etc.) can maintain the pseudo capacitance property almost under any circumstances, while the pseudo capacitance performance of more materials needs to create conditions: some surface controlled reactions cannot be replaced by larger ions; some typical battery materials can also behave like pseudocapacitors as long as they are "nano sized" enough to become "surface everywhere". Secondly, the scanning speed is also a key factor: when the scanning speed is fast enough, the surface reaction will have "peak separation". Because the surface reaction is not a simple physical adsorption desorption process after all, it cannot reach the same speed as the double layer capacitors (EDLCs), and the gap between EDLCs and surface reaction is widened under the large scanning speed, becoming "less like capacitors". Therefore, I think it is feasible to use "capacitor (F)" or "battery (Ah)".

Round 2
Reviewer 1 Report
Page 5, line 127:….. capacitance or capacity?
Figure 6 The impedance plot is not adequately discussed. For this, it is necessary to deepen the discussion and support it with the literature, otherwise it would have to be eliminated from the figure since it does not contribute anything to the discussion.

Author Response
Response Letter to Editor & Reviewers
9 November 2022
Dear Editor & Reviewers:
Thanks again for the efforts to enable the further quality improvement on this manuscript. The point-by-point response can be found below.
Reviewer#1: Page 5, line 127:….. capacitance or capacity?
Response/Revision: Thank you for your comments. This has been modified, see the green part in line 127 of the text.
Figure 6 The impedance plot is not adequately discussed. For this, it is necessary to deepen the discussion and support it with the literature, otherwise it would have to be eliminated from the figure since it does not contribute anything to the discussion.
Response/Revision: Thank you for your comments. We added the electrochemical impedance diagram of the sample after 1000 charge discharge cycles, compared the test diagram before and after the cycle, and redrawn Figure 6d. At the same time, the interpretation of the main body has been rewritten. See the green part of the main body on page 7 of the main body for the revised content.
Reviewer 2 Report
The paper can be accepted without any further changes.
Reviewer 3 Report
I appreciate the Authors’s answers to my comments and questions. The quality of the manuscript is now certainly improved. I must insist, however, to add all calculations procedures used to evaluate the electrochemical performance of materials in electrochemical capacitors (both in a two- and a three-electrode configuration). It is missing whilst proper performance metrics are crucial for the true evaluation of electrical parameters of faradaic-type materials with battery-like characteristics (such as in the revised paper). I propose using suggestions given in papers I previously advised in case the current calculations are done inappropriately. The total mass loading in produced electrodes (including binder and conductive agent) should be given as well in the experimental section.
Author Response
Response Letter to Editor & Reviewers
9 November 2022
Dear Editor & Reviewers:
Thanks again for the efforts to enable the further quality improvement on this manuscript. The point-by-point response can be found below.
Reviewer#3: I appreciate the Authors’s answers to my comments and questions. The quality of the manuscript is now certainly improved. I must insist, however, to add all calculations procedures used to evaluate the electrochemical performance of materials in electrochemical capacitors (both in a two- and a three-electrode configuration). It is missing whilst proper performance metrics are crucial for the true evaluation of electrical parameters of faradaic-type materials with battery-like characteristics (such as in the revised paper). I propose using suggestions given in papers I previously advised in case the current calculations are done inappropriately. The total mass loading in produced electrodes (including binder and conductive agent) should be given as well in the experimental section.
Response/Revision: Thank you for your comments. Let me briefly describe the calculation process:
According to formula (where I, t, ΔV and A are discharge current (A), discharge time (s) , potential window respectively(ΔV) and electrode area (cm2)) can be calculated in the two or three electrode system under different current densities. For the papers you suggested last time, we also added them to the body of the paper in the form of references 28 and 30. Thanks again for your recommendation.
For the total mass of the load on the produced electrodes, we have described it in section 3.3 of the text: the active substance CoSn(OH)6, carbon black and PTFE are mixed at a mass ratio of 8:1:1, and then pressed to 1 × 1 cm2 of foam nickel. The total mass of the final substance load is 3.1 mg cm2 (the description of this part is marked in green in the text).